# What helps and hinders doctors in engaging in continuous professional development? An explanatory sequential design

**Carrie H. K. Yam**  *, **Sian M. Griffiths, Eng-Kiong Yeoh**

The Jockey Club School of Public Health and Primary Care, The Chinese University of Hong Kong, Hong Kong SAR, PR China

* carrieyam@cuhk.edu.hk

**Data Availability Statement:** All relevant data are within the manuscript and its Supporting Information files.

## Abstract

### Background

Licensure and registration are the traditional approaches to ensure minimally acceptable standards of care for practice. However, due to advances in medical technology and changes in clinical practice, the knowledge and skills obtained from basic education and training may rapidly become out of date. There is no mandated, structured and ongoing mechanism to assess all doctors' competence in Hong Kong. This paper assessed doctors' perceived needs for continuous professional development, and to identify facilitators and barriers that are likely to influence the implementation of compulsory continuous professional development for maintaining professional competence and ensuring patient safety.

### Methods

An explanatory sequential mixed method design with two distinct interactive phases was adopted comprising a postal self-administered questionnaire survey among a random sample of 2,459 of doctors (Phase 1), followed by individual interviews of a stratified sample of 30 questionnaire respondents for the subsequent qualitative analysis (Phase 2).

### Results

The majority of doctors (over 90%) agreed the importance of continuous professional development to update knowledge and skills. However, just 30.7% of non-specialists compared with 65.4% of specialists agreed it would be desirable for continuous professional development to be a requirement for renewal of licenses. A relatively higher percentage of non-specialists compared with specialists reported barriers to participation such as accessibility, availability and relevance of the content of the programmes. Facilitators for uptake included more convenient schedule and location, relevant content, and incentives for participation such as making this a pre-condition for enrolling in government-funded services.

### Conclusions

To address the needs of individual doctors, the spheres of practice, personal preferences and learning styles should be considered in deciding the content and processes of

**Funding:** This work was supported by the Health and Medical Research Fund of the Food and Health Bureau of the Hong Kong Special Administrative Region [RFW-CUHK]. The funder had no role in study design, data collection and analysis, decision to publish, or preparation of the manuscript.

**Competing interests:** The authors declare that they have no competing interests.

continuous professional development. Flexibility is also an important principle. A learning model, incentives for participation and a compliance strategy (instead of deterrence) could be effective strategy for continuous professional development.

## Introduction

The fundamental purpose of professional regulation is to ensure minimally acceptable standards of practice and care to maintain public confidence in the safety and quality of care provided [1]. It also aims to improve the quality of care, provide guidance about best practices and continuous professional development, ensure fitness to practice, and foster performance improvement through continuous measurement and feedback processes. Licensure and registration are the traditional approaches and mechanisms to ensure minimally acceptable standards of care [1]. However, due to the advances in medical knowledge and technology and changes in clinical practice, the knowledge and skills obtained from basic medical education and training rapidly become outdated. A systematic review of 62 studies investigating the association of experience of doctors and quality of care found around half of the studies reporting a decrease in performance with increasing years of practice [2]. Contemporary healthcare professionals are expected to participate in continuous professional development seen as a minimum requirement to demonstrate their knowledge and skills are continuously updated, beyond basic requirements for practice at entry to the profession.

Regulation of healthcare professionals exists in two forms, each with a different locus of control and intended outcome [3]. The first form ensures minimally acceptable standards of practice and to identify and deal with potentially poorly performing doctors, backed by sanctions, which is normally exercised by external regulatory bodies through *"command and control"* mechanisms of influence. The second form serves to guide professional development and help all doctors improve, influencing practice through financial or non-financial *"incentives"*. It is intended to promote life-long learning. As it is controlled by the individual and the profession, it is perceived as non-threatening. There are also two approaches in the regulation of doctors: the assessment model and the learning model which serve two different purposes complementing each other and maybe used together to ensure continuing competence of doctors [4]. The assessment model emphasizes standards and performance, and includes different types of measurements e.g. interviews, record reviews, oral examinations in case-based format, patient satisfaction questionnaires and ratings from peers to detect poor performance and adjudication of professional misconduct by the regulatory body for control. It sets standards that must be reached. It starts by requiring minimum standards through registration and licensing, and then moves to assessing doctors who have been the subject of complaints, and may include periodic assessment and screening of all doctors or of those doctors who are categorized as at risk of meeting standards. Revalidation and recertification are examples of the assessment model developing in many jurisdictions including United Kingdom (UK) and United States of America (US) to continuously monitor doctors' performance to ensure patient safety [5]. There is another model, the learning model, which seeks to improve clinical competence but does not identify doctors who perform poorly; by which health professionals keep updated to meet the needs of patients, health services, and their own professional development [6]. The activities involve continuous professional development activities, self-assessment of learning needs, academic activities and audits that focus on continuous quality improvement. Traditionally, Continuing Medical Education (CME) activities—in the form of formal lectures or seminars with time-based credit points awarded for the activity–are used to

update knowledge and maintain competence. However, CME is viewed as rather "passive" learning, and should be seen as a minimum requirement for continuing competence. Internationally, there is a move from CME (or clinical update) to Continuous Professional Development (CPD), which is a relatively active self-directed learning including medical, managerial, social and personal skills [6]. This learning model is permissive and supportive.

Medical doctors in Hong Kong have a high degree of self-regulation. The healthcare system, including regulation of the profession, is dominated by the medical profession. To become a medical doctor in Hong Kong, one must enroll in a 6-year medical programme provided by two medical schools in Hong Kong. On completion of the medical degree, graduates are given provisional registration by the Medical Council of Hong Kong (MCHK), statutory authority for regulation of doctors, to train as interns in a public hospital operated by the public sector corporation, the Hong Kong Hospital Authority. After being assessed as having satisfactorily completed the 12-month internship training, full registration is given by the MCHK and the doctor may practice medicine independently. Many will however choose to apply for specialist training with a standard 6-year format for basic and higher specialist training overseen by another statutory body, the Hong Kong Academy of Medicine and its Colleges. However, it is possible to go into general practice as a non-specialist without any further professional training or development. According to a health manpower survey by Department of Health, 66.7% of all doctors in Hong Kong work as specialist (family medicine is an accredited specialist practice), while 29.2% are in general practice [7]. Nearly half (48.9%) of all doctors are working in the private sector, while 46.9% in the public sector. Doctors have relatively a high social status and are paid well in Hong Kong. The salary of a resident in the public sector ranges from US $8,700–18,000 per month, which is much higher than the median monthly wage of Hong Kong employees (US$2,200) [8]. The salary of doctors in private sector is in general higher than that of public sector, particularly for those in specialist practice. Though doctors have relatively higher income in Hong Kong, there are concerns about the long working hours and excess workload in public hospitals particularly during the influenza season [9].

To guide professional development and help all doctors improve, healthcare professionals are expected to participate in continuous professional development to update their knowledge and skills in a learning model. However, there is no mandated, structured and ongoing mechanism in Hong Kong to assess all doctors' performance and ensure competence and improve the quality of care [10]. Only specialists are mandated to participate in continuous professional development (named as "CME CPD programme" which is differentiated into components of CME and CPD with guidelines on the types and hours of programmes) operated by the Hong Kong Academy of Medicine to meet the requirements for accreditation as specialists. However, neither CME nor CPD is mandated for renewal of practicing certificates for licenses by the MCHK. Non-specialists undertake CME on a voluntary basis. To incentivize uptake of CME, the MCHK has a CME programme for non-specialists which issues "CME-Certified" status to those doctors who have fulfilled the CME requirements. However, only around 20% of non-specialists obtained "CME-Certified" status in 2017. Currently, complaints from public/ patient is the main channels for detecting poor performance and professional misconduct of doctors, and is reactive. If the doctor is found guilty of misconduct in a professional respect, MCHK will issue a range of disciplinary sanctions: the most serious of which is removal from the Register of licensed practitioners, suspension from practice for a defined period not exceeding 3 years, a reprimand or issue of a warning letter. In 2017, of the 26 disciplinary inquiries conducted by the MCHK, 18 cases were found to be guilty of professional misconduct: 7 were disciplined with suspension due to their disregard of professional responsibility to patients, 9 for conviction in court arising from failure to keep a register of dangerous drugs or dangerous driving, and 2 for practice promotion/ quotable qualification [11]. Inadequacies in

the assessment model to ensure competence and fitness to practice could constitute to medical errors and could be resulted in serious injury or death of the patient. In 2017–18, the incident rate of sentinel events was 1.1 cases per one million patient appointments in public hospital in Hong Kong, down from 2.7 cases per one million patient appointments ten years ago [12]. However, the increasing cost of medical insurance and litigation might cause heavy financial burden to the medical profession and government. Though it might be unrealistic to expect zero mistakes in any healthcare system, sentinel events can be minimized by ensuring competence of healthcare professionals in combination with other institutional safeguards. There is a global trend for a mandatory requirement for continuous professional development for all healthcare professionals including doctors, nurses, and dentists etc. in order to maintain professional competence [5]. Participation in continuous professional development is considered the minimum mechanism needed to uphold standards of professional competence and conduct in order to ensure and enhance patient safety and patient-centred care.

This study assessed doctors' perceived needs for continuing professional development, and to identify factors that are likely to influence the implementation of compulsory continuing professional development including barriers and facilitators to changing the current system for maintaining professional development and ensuring patient safety.

## Methods

An explanatory sequential mixed method design was adopted. There were two distinct interactive phases i.e. an initial quantitative data collection and analysis was conducted among the doctors (Phase 1), followed by individual interviews of a random sample of respondents who had completed the questionnaire survey for subsequent qualitative data collection and analysis (Phase 2). The explanatory design was chosen since the subsequent qualitative data would help in explaining the initial quantitative results and provide more in-depth understanding on the research questions [13]. Ethics approval was obtained for the study from the Survey and Behavioral Research Ethics Committee at the Chinese University of Hong Kong. Written consent was obtained from all respondents after a clear explanation of the study objectives and to ensure data confidentiality.

### Quantitative phase 1 –Postal self-administered questionnaire survey

In the first quantitative phase of the study, a postal self-administered questionnaire survey was conducted among doctors to obtain a picture of their needs for continuous professional development and the barriers encountered. The target population was the 12,688 medical doctors who have full registration (resident list only) in the most recent "List of Registered Doctors" of the Medical Council of Hong Kong on 23 February 2015. In order to achieve a precision level of plus/minus 5% from the true value at 5% significant level and 80% power with the conservative assumption of 50% of respondents perceiving the importance of continuous professional development, the target sample size was 737 doctors for the questionnaire survey. Assuming a 30% response rate, a random sample of 2,457 doctors was drawn for the postal questionnaire survey. To facilitate the return of completed questionnaires, the questionnaires were sent together with a covering letter which explained the aim of the study and the assurance of confidentiality and privacy. A prepaid and self-addressed envelope was also enclosed. Up to two reminders were made for non-respondents. The first reminder together with the questionnaire was sent again to those who had not replied after 14 days (i.e. the second mailing). Similarly, a second reminder (i.e. the third mailing) was sent out after another two weeks.

The questions were drafted taking reference from the literature of studies of doctors' views towards medical regulation and continuous professional development [14–16]. The

questionnaire was reviewed and commented on and revised by experts, and pilot-tested before implementation in the main study. The questionnaire focused on the perceived needs for and barriers to continuous professional development (S1 File). There were four questions relating to the life-long learning needs e.g. whether they recognized the need to regularly update medical knowledge. After understanding their needs, we asked whether the respondents participated in any self-learning and CME CPD programmes in the past one year, and what if any barriers were encountered. We further explored their perception on the usefulness of continuous professional development e.g. whether taking part in CME CPD programmes can improve their practical skills or patient outcome; and whether there should be a compulsory requirement for participation in CME CPD programmes. The questions were presented in a 4-point Likert scale showing the agreement of the participants towards the questions. Pilot testing was conducted among 10 medical doctors through face-to-face interviews and minor amendments were made to the text of the questionnaire based on the feedback received.

## Qualitative phase 2 –Individual semi-structured telephone interview

In order to better understand the needs of all medical profession for continuous professional development and how this can be better met, the second qualitative phase was conducted through individual semi-structured telephone interviews among a pool of respondents as a follow up to the quantitative results.

The results of the quantitative phase were firstly analyzed to identify the key variables which accounted for the differences in the doctors' perceived needs for continuous professional development. This enabled the identification of doctors with different perceived needs and attitudes towards CME CPD programmes for the semi-structured telephone interview to further understand their different needs and attitudes. We stratified the respondents according to

i. Whether respondents agreed or disagreed *"All doctors in HK should participate in CME CPD programmes recognized by Medical Council of Hong Kong/ Hong Kong Academy of Medicine"*,

ii. Whether they agreed or disagreed *"CME CPD should be required for all doctors in Hong Kong for renewal of practicing certificates"*,

iii. Age of respondents, and

iv. Whether doctor is a specialist or non-specialist.

In this second qualitative phase, 30 respondents were selected using a stratified random sampling method. Qualitative studies generally requires 20–30 interviews to reach saturation of data [17]. A proportional sampling framework was used with a minimum sample size requirement in each of the categories of doctors delineated by the criteria above. The data collection method was an individual semi-structured telephone interview using a discussion guide which was developed from the quantitative results. An invitation letter was sent to each respondent to invite them for interview at a date and time convenient to them. Each individual interview (lasting approximately 15–20 minutes) was conducted by two persons, one leading the interview and the other taking notes. The proceedings were audio-recorded with the participants' consent. Interviewees were allowed to freely express their views regarding the discussion topic. A common protocol was developed and followed for all individual interviewees to achieve consistent and optimal results. In order to ensure an in-depth examination of the topic, a set of open-ended questions (and probe for responses) were used to further understand their perceived needs for and barriers to continuous professional development based on the

responses from the participants (S2 File). For example, for those who did not agree that all doctors should participate in CME CPD programmes, we explored their reasons why not, and if they had any suggestions for doctors to keep medical knowledge updated. We also asked participants their views on how to reduce barriers to CME CPD participation, and ways to encourage doctors' participation. The preferred type of activity and format of CME CPD programmes were also explored.

For questionnaire survey, data entry was started after receiving the first round of responses, with on-going data validation using preset values. Both descriptive and multivariate analyses were conducted using SPSS 16.0. First, descriptive statistics were conducted both for the whole sample, and for different doctor profile variables such as whether the doctor is a specialist or non-specialist. Distribution of each measure was presented. Continuous data was presented as means ± standard deviations. Percentages were calculated for dichotomous variables. Univariate analysis was performed by Pearson's Chi squared test, Student's t-test, or the Mann-Whitney U test, as appropriate. Factors with P-values of <0.05 were then entered into a multivariate logistic regression model to examine their independent effects after adjusting for the other factors. All P-values were two-sided, and <0.05 were reported as statistically significant.

For individual semi-structural telephone interview, the transcripts were analyzed for recurring themes with the aid of NVivo software. A thematic analysis was adopted. The transcript was read by on researcher to identify possible broad themes. If these emergent themes occurred repeatedly across and within the transcripts, they were noted as recurrent themes. Similarly, a second researcher read the transcripts and generated emergent and recurrent themes independently. Subsequently, the two researchers discussed and agreed on emergent themes, then examined the transcripts for any connections among the recurrent themes. Researchers discussed with the principal investigator to agree on the themes. Related recurrent themes were grouped under a master theme. Extracts from the transcripts were used to interpret the themes.

## Results

A total of 2,459 questionnaires were sent in March 2015 to a pool of doctors randomly selected from the MCHK list of fully registered doctors. After 3 mailings, 870 questionnaires were returned by July 2015 with a response rate of 35.4%. The profile of doctors was shown in Table 1. 62.2% were specialists and 37.8% were non-specialists. The non-specialists were relatively younger (55.3% aged 21–40) compared to specialists (34.6%). Specialists were more likely to work in the public hospitals (56.4%) as compared to non-specialists (41.6%).

Table 2 summarized the four key themes identified from the questionnaire survey and focus group discussions: (i) perceived needs for continuous professional development, (ii) participation in CME CPD programmes, (iii) barriers to CME CPD participation, and (v) facilitators of participation in CME CPD programmes.

### (i) High perceived needs for continuous professional development

There was a high degree of perceived needs for continuous professional development (Table 3). The majority (99.2%) expressed a need to be updated on knowledge and for development of new clinical skills due to advances in medical knowledge and technology. 98.7% recognized their own need to regularly update medical knowledge, and 97.9% agreed that all practicing doctors needed to have their professional knowledge updated. A significant proportion (95.2%) agreed with the statement "I will fall behind in standards of my professional practice if I stopped learning about new developments". Most of the doctors recognized the

**Table 1. Respondents' profile.**

| | Specialist (n = 541) | Non-specialist (n = 329) | All doctors (N = 870) | P-value |
|---|---|---|---|---|
| **Gender (Male)** % | 71.3 | 64.1 | 68.6 | 0.026 |
| **Age %** | | | | |
| 21–40 | 34.6 | 55.3 | 42.4 | 0.000 |
| 41–60 | 49.2 | 24.0 | 39.7 | |
| 61 or above | 16.3 | 20.7 | 17.9 | |
| **Places of first degree** % | | | | |
| Hong Kong | 86.3 | 76.6 | 82.6 | 0.000 |
| Overseas | 13.7 | 23.4 | 17.4 | |
| **Main setting of current practice %** *(multiple options allowed)* | | | | |
| Hospital Authority | 56.4 | 41.6 | 50.8 | 0.000 |
| Government | 5.0 | 8.5 | 6.3 | 0.039 |
| Private Hospital | 7.6 | 3.6 | 6.1 | 0.019 |
| Academic Institution | 5.7 | 4.0 | 5.1 | 0.246 |
| Solo private practice | 21.1 | 25.5 | 22.8 | 0.128 |
| Group private practice | 7.0 | 14.3 | 9.8 | 0.000 |
| **College(s) registered (among Specialists)** *(multiple options allowed)* | | | | |
| Anaesthesiologists | 6.3 | - | - | - |
| Community Medicine | 3.0 | - | - | - |
| Emergency Medicine | 5.0 | - | - | - |
| Family Physicians | 10.0 | - | - | - |
| Obstetricians & Gynaecologists | 9.1 | - | - | - |
| Ophthalmologists | 3.7 | - | - | - |
| Orthopaedic Surgeons | 5.5 | - | - | - |
| Otorhinolaryngologists | 1.7 | - | - | - |
| Paediatricians | 7.4 | - | - | - |
| Pathologists | 4.3 | - | - | - |
| Physicians | 24.2 | - | - | - |
| Psychiatrists | 5.2 | - | - | - |
| Radiologists | 5.9 | - | - | - |
| Surgeons | 9.1 | - | - | - |

effectiveness of continuous professional development, agreeing that the participation in CME CPD activities can improve practical skills (85.1%) and improve patient outcome (72.9%) (Table 4). The agreement was relatively higher for specialists relative to non-specialists.

The individual interviews echoed similar results. The majority of the interviewees (in particular for specialists) agreed CME CPD programmes could help update knowledge, improve patient outcomes and practical skills; but the effectiveness depended on the format and whether it was passive or active:

> "*I felt that if I don't catch up for a few months, things are already different. The medications used nowadays are completely different from those used five or six years ago. For example, some medications used years ago might have more side effects, but medications that used currently have fewer side effects. So, I think if you are not able to keep yourself up to date, it is difficult to practice medicine safely.*" *(A specialist working in the public sector)*

Among all the types of activities, doctors expressed that reading journals was the easiest and most convenient way to help in updating knowledge. Attending lectures and conferences

**Table 2. Summary of themes.**

| Theme | Questionnaire survey | Individual interviews |
|---|---|---|
| (i) High perceived needs for continuous professional development | • Agreed that<br>• Advances in medical knowledge and technology require update of knowledge (99.2%)<br>• My need to regularly update medical knowledge (98.7%)<br>• All practicing doctors need to keep their professional knowledge updated (97.9%)<br>• I will fall behind in standard of my professional practice if I stopped learning about new developments" (95.2%)<br>• Agreed that<br>• The participation in CME CPD activities can improve practical skills (85.1%) and improve patient outcome (72.9%) | CME CPD can help to update knowledge, improve patient outcome and practical skills; but the level of effectiveness depends on its format and types of activities i.e. passive or active |
| (ii) Participation in CME CPD programmes | • 74.3% agreed all doctors are required to participate in CME CPD programmes<br>• 52.3% agreed CME CPD should be required for renewal of practicing certificates.<br>• 47.2% agreed CME CPD should be included as one of the criteria for joining the government healthcare programmes. | • *Reasons for agreement*:<br>• International trend and the basis for patient safety<br>• Belief that doctors' self-discipline was not sufficient<br>• *Reasons for disagreement*:<br>• Practical limitation of obtaining sufficient credit points e.g. time, location, no vacancy, activities and/ or topics not relevant in particular for non-specialists<br>• High resistance due to the belief that doctors had already fulfilled the basic qualification to practice after graduation; experience can be gained through daily clinical practice |
| (iii) Barriers to CME CPD participation | • 59.6% had barriers to CME CPD participation<br>• Of the barriers encountered, it was mainly related to time (62.5%), followed by work-life balance (45.1%), inconvenience of the CME CPD activities (34.8%), cost (17.3%) and unavailability of suitable activities (10.5%) | • Accessibility and availability<br>• Coverage and topics of CME CPD |
| (iv) Facilitators of participation in CME CPD programmes | NA | • Operational/ logistics:<br>• Flexible CME CPD<br>• Centralized CME calendar<br>• Content:<br>• Variety of content<br>• Sufficient platform<br>• Related to current issues and daily routine practice<br>• Policy:<br>• Balance between work and life<br>• More education<br>• Case allowance/ tax reduction<br>• Public-Private-Partnership as a lever |

could also enhance knowledge, but was subject to the quality. Audit was perceived to be the most useful activity in improving practice; while case presentation and peer review could help improve patient outcomes through peer communication and experience sharing, and facilitate them to update clinical protocols, thereby resulting in improvement in practice:

## (ii) Participation in CME CPD programmes

Table 3 showed the percentage of agreement with participation in CME CPD programmes. 74.3% of doctors agreed that all doctors should participate in CME CPD programmes (82.1% for specialists and 61.4% for non-specialists). However, only around half (52.3%) thought it should be a requirement for renewal of practicing certificates. Non-specialists were significantly more likely to disagree with the requirement of CME CPD programmes as a condition for renewing practicing certificates with only 30.7% of non-specialists agreeing compared with 65.4% of specialists who agreed. Slightly less than half of all doctors (47.2%) agreed that CME CPD programmes should be required as one of the criteria for joining any government subsidized healthcare programmes.

Table 3. Perceived needs for continuous professional development.

| | Specialist (n = 541) | Non-specialist (n = 329) | All doctors (N = 870) | P-value |
|---|---|---|---|---|
| **Advances in medical knowledge and technology require updating of knowledge and development of new skills for medical professionals.** | | | | |
| Strongly agree | 56.9 | 43.5 | 51.8 | 0.002 |
| Agree | 42.6 | 55.3 | 47.4 | |
| Disagree | 0.4 | 0.6 | 0.5 | |
| Strongly disagree | 0.2 | 0.6 | 0.3 | |
| **I recognize my need to regularly update my medical knowledge.** | | | | |
| Strongly agree | 48.8 | 37.4 | 44.5 | 0.000 |
| Agree | 50.8 | 59.9 | 54.3 | |
| Disagree | 0 | 1.2 | 0.5 | |
| Strongly disagree | 0.4 | 0.6 | 0.5 | |
| Don't know | 0 | 0.9 | 0.3 | |
| **I will fall behind in standards of my professional practice if I stopped learning about new developments in my specialty/ area of practice.** | | | | |
| Strongly agree | 40.2 | 30.4 | 36.5 | 0.007 |
| Agree | 56.3 | 62.6 | 58.7 | |
| Disagree | 2.0 | 4.9 | 3.1 | |
| Strongly disagree | 1.1 | 0.9 | 1.0 | |
| Don't know | 0.4 | 1.2 | 0.7 | |
| **All practicing doctors need to keep their professional knowledge updated.** | | | | |
| Strongly agree | 44.5 | 29.5 | 38.8 | 0.000 |
| Agree | 54.2 | 67.2 | 59.1 | |
| Disagree | 0.6 | 1.5 | 0.9 | |
| Strongly disagree | 0.4 | 0.3 | 0.3 | |
| Don't know | 0.4 | 1.5 | 0.8 | |

Adjusted for gender and age, logistics regression showed that those who were aged between 41–60 relative to those aged 21–40 (OR: 1.46; CI: 1.01–2.13) and specialist (OR: 3.96; CI: 2.87–5.46) were more likely to agree that CME CPD programmes should be required for all doctors for renewal of practicing certificates. Being a specialist (OR: 2.62; CI: 1.91–3.57) was the only predictor of agreeing that CME CPD programmes should be included as one of the criteria for joining government-funded healthcare programmes.

The individual interviews specifically explored the views on the requirement of compulsory CME CPD programmes as a pre-condition for renewal of practicing certificates. The reason for the majority of those who agreed was because it was an international trend and it was a basis for patient safety:

> "*Anyway, I think it's an international trend (for compulsory CME CPD) because medical advancement is really fast. Technology and the medical profession is changing rapidly.*" *(A non-specialist working in the public sector)*

Some further pointed out that the belief "doctors are driven by self-discipline to participate in CME CPD programmes" was insufficient to ensure doctors' participation. Only compulsory CME CPD requirement would force all doctors to participate the CME CPD programmes:

> "*(Compulsory CME CPD) is a quality assurance. Medical information is changing all the time—if you do not update your knowledge, you will lag behind. So, I hope all doctors can achieve this. . . . . . it is difficult for the general public to trust that you will take part in CME*

**Table 4. Attitudes towards continuous professional development.**

| | Specialist (n = 541) | Non-specialist (n = 329) | All doctors (N = 870) | P-value |
|---|---|---|---|---|
| **Generally, participation in CME CPD activities (which is recognized by MCHK/ HKAM) can improve practical skills.** | | | | |
| Strongly agree | 12.4 | 9.4 | 11.3 | 0.008 |
| Agree | 75.0 | 71.7 | 73.8 | |
| Disagree | 8.5 | 8.8 | 8.6 | |
| Strongly disagree | 0.7 | 1.2 | 0.9 | |
| Don't know | 3.3 | 8.8 | 5.4 | |
| **Generally, participation in CME CPD activities (which is recognized by MCHK/ HKAM) can improve patient outcome.** | | | | |
| Strongly agree | 8.1 | 5.8 | 7.2 | 0.065 |
| Agree | 68.1 | 61.7 | 65.7 | |
| Disagree | 9.1 | 11.9 | 10.1 | |
| Strongly disagree | 1.1 | 1.5 | 1.3 | |
| Don't know | 13.5 | 19.1 | 15.7 | |
| **All doctors in Hong Kong should participate in CME CPD programme organized by MCHK/ HKAM.** | | | | |
| Strongly agree | 19.0 | 9.1 | 15.3 | 0.000 |
| Agree | 63.0 | 52.3 | 59.0 | |
| Disagree | 10.9 | 21.0 | 14.7 | |
| Strongly disagree | 1.7 | 5.2 | 3.0 | |
| Don't know | 5.4 | 12.5 | 8.0 | |
| **CME CPD should be required for all doctors in Hong Kong for renewal of practicing certificates.** | | | | |
| Strongly agree | 13.7 | 3.0 | 9.7 | 0.000 |
| Agree | 51.8 | 27.7 | 42.6 | |
| Disagree | 23.7 | 42.9 | 30.9 | |
| Strongly disagree | 3.9 | 15.8 | 8.4 | |
| Don't know | 7.0 | 10.6 | 8.4 | |
| **CME CPD should be included as one of the criteria for joining government-initiated healthcare programmes e.g. vaccination programmes, elderly healthcare vouchers.** | | | | |
| Strongly agree | 7.8 | 4.6 | 6.6 | 0.000 |
| Agree | 47.3 | 29.8 | 40.7 | |
| Disagree | 28.8 | 40.4 | 33.2 | |
| Strongly disagree | 3.7 | 12.2 | 6.9 | |
| Don't know | 12.4 | 13.1 | 12.6 | |

* MCHK refers to Medical Council of Hong Kong and HKAM refers to Hong Kong Academy of Medicine.

> CPD based on your self-discipline......If CME CPD is not compulsory, it could only encourage some self-driven doctors who are already interested in continuous learning to take part, but you cannot change those doctors who are not self-driven." (A specialist in solo private practice)

The doctors who did not agree with the compulsory CME CPD requirement expressed their concerns about the practical limitations in obtaining sufficient credit points for renewal of practicing certificates, arising from the inflexibility of CME CPD activities. The encountered barriers included: the scheduled time was not convenient, location problems, the programmes were fully subscribed, activities and/ or topics/ content were sometimes not relevant, in particular for non-specialists. They also had the concerns that a few doctors would just sign the attendance of the seminars/ conferences to earn credit points (to fulfill the requirement) and then leave without gaining knowledge/ skills from the activities:

"*I think compulsory CME CPD is not feasible. . . . . . However, for me to get enough credits, I need to take at least ten classes. But, these are time clashes with my schedule, or I am not interested in all of the topics. Therefore, I cannot obtain enough credits. . . . . .On the other hand, though compulsory CME CPD could force doctors to go to seminars in order to get sufficient credits, they might not listen attentively. Therefore, requiring CME CPD compulsory may not necessarily enable doctors to improve.*" *(A specialist working in the public sector)*

Some of the doctors expected that there would be a high degree of resistance from the profession if CME CPD programmes were made a compulsory requirement since doctors thought that they already achieved the required qualification to practice on entry to the profession. They believed encouraging voluntary participation would be better than requiring compulsory participation with sanctions for not doing:

"*Doctors believe that they have already graduated, so they already have the standard to practise. If you make additional requirements i.e. compulsory for all doctors to take part in CME CPD to renew their license, doctors will resist this decision. They think only specialist should fulfil the requirements for CME CPD by the Hong Kong Academy of Medicine to retain their higher specialist qualification. . . . . . This is a historical problem–in the past there has been discussion for compulsory CME CPD, however it was stopped because of high degree of resistance from doctors. In my opinion, I see there is still a high degree of resistance.*" *(A specialist in solo private practice)*

"*I feel that voluntary participation or encouragement to participate will be better, because we (doctors) are already qualified after completing the five-year medical course. So I feel that there is no need to participate CME CPD for license renewal.*" *(A non-specialist in a private hospital)*

In addition, some doctors thought it was their own duty (self-discipline) to study (depending on their own needs) based on their own code of practice. Therefore, compulsory CME CPD requirement is not needed. Some further thought that they could gain enough experience through daily clinical practice; therefore, they did not need compulsory attendance at CME CPD programmes to update knowledge.

"*Actually, the medical profession has the oath (code of conduct) to provide the best quality service which is patient-centre care. This oath would motivate doctors to update knowledge voluntarily.*" *(A specialist in a private hospital)*

### (iii) Barriers to CME CPD participation

35.7% of all doctors said they did not encounter any barriers to CME CPD participation. 42.9% encountered "few" barriers whereas 16.7% encountered "significant/a great deal of" barriers. Non-specialists (65.4) were significantly more likely to encounter barriers as compared with specialists (56.2%). Of the barriers encountered, 62.5% were mainly related to time, followed by 45.1% related to work-life balance. Inconvenience of the CME CPD activities accounted for 34.8%, cost for 17.3%, and unavailability of suitable activities for 10.5%.

The individual interviews were able to further elaborate on the nature of the barriers encountered in the context of (i) accessibility and availability of, and (ii) coverage and topics of CME CPD programmes. In terms of accessibility and availability, the main concern was about time issues due to their heavy workload, and the need for work-life balance due to family commitment during weekends:

"*Recently, I felt that the amount of work is increasing and I am getting busier, and that really affected the quality of my CME CPD participation [audit or peer review] because of the limited amount of time to prepare. For example, it takes time to discuss a case with colleagues, to prepare and present a topic, or to listen to a colleague's presentation. Busy schedule and work affects my focus in working and preparation of these activities. . . . . . . I want to keep doing a better job but I am really busy, this is a major barrier.*" *(A specialist working in the public sector)*

"*If it [CME activity] is held at my hospital after work, that is okay. However some activities usually take place outside my hospital during the period which I have to work, or on Saturday or Sunday when I may have family commitments, so the timing may be difficult.*" *(A specialist working in the public sector)*

Another practical issue was location in particular for doctors in the private sector. Also, some doctors expressed the lack of a centralized system for the CME CPD activities organized by different providers hindered their application since they need to check CME CPD activities from different sources.

In terms of coverage and topics of CME CPD programmes, most doctors expressed concerns about the topics with CME CPD programmes which may not be relevant and some of them were not interesting, which discouraged them from attending. A few doctors also mentioned about the cost issues relating to some of the lectures. It was relatively easy for doctors working in the public sector to participate in CME CPD activities e.g. journal clubs and seminars at their hospitals. There were concerns of the difficulty for doctors in the private sector in particular for those who were working in clinics due to their work nature of individual practice because they were very busy. Therefore, they suggested on-line learning for doctors in the private sector.

### (iv) Facilitators of participation in CME CPD programmes

In order to facilitate the uptake of CME CPD programmes, doctors suggested improvement in three aspects of the programmes: operational and logistics, content, and policies.

### Operational and logistics aspect

Flexible time and location of CME CPD activities would be able to facilitate doctors' participation. A centralized CME CPD calendar would make it convenient for doctors to access information of the programmes that are provided. A review of application procedures for both recognizing (e.g. peer review, shorter application time) and registration of courses was also recommended:

"*I personally believe that having flexibility is better, because most doctors are very busy–they need to do lots of things taking care of patients. Flexibility could include having no fixed format of CME CPD activities and not necessarily having to physically attend the activities. For example, they can self-study through an online platform rather than going to seminars. This is more flexible, and through self-learning they can also keep their knowledge up-to-date.*" *(A specialist in group private practice)*

### Content aspect

The doctors suggested that if CME CPD course could be more related to current issues and routine practice e.g. influenza in winter surge, updated screening information e.g. colorectal

cancer screening, it would encourage doctors to enroll in the programmes. A few proposed to extend the coverage of topics e.g. communication skills or humanities:

> "*The relevance of the CME CPD activities is the most important factor, for example now some of the CME CPD courses are about the new SARS, Middle East Respiratory Syndrome which is very relevant to us. . . . . . Maybe CME CPD should have components of General Studies. . . . . . In recent years, TV advertisements promoted that Hepatitis B can be treated, and also to promote colorectal cancer screening. Because of these advertisements, more people will go to the doctors to ask for more information. I think it will be useful to develop a CME CPD activity on colorectal cancer screening.*" *(A non-specialist in a private hospital)*

Specifically, to encourage non-specialists to participate, some specific measures were proposed: a lower requirement of credit points for non-specialists; sufficient platform to get the credit points e.g. online courses for non-specialists; relevant courses for non-specialists e.g. legal cases sharing:

## Policy aspect

Some doctors suggested a cash allowance/ tax reduction could be considered to encourage participation. Specially for non-specialists, a public-private-partnership (PPP) could be a lever for change e.g. government PPP programmes e.g. chronic diseases programmes and elderly healthcare voucher to be linked with CME CPD requirements. The government could also consider requiring the private sector to enroll doctors who have participated in CME CPD programmes if they want to join any PPP programmes, to encourage participation and improve quality. However, there were still practical concerns e.g. quality of content and the participation rates in PPP programmes:

> "*Through these PPP programmes (with financial incentives), it will attract doctors to attend CME CPD activities. Doctors will feel that there is an opportunity for continuous learning to gain more knowledge while in return they also earn more income.* I think this is a good way to encourage doctors to attend these activities.*" *(A non-specialist in solo private practice)*

Some thought it was better for CME CPD programmes to be voluntary and from the perspective of encouragement and basis of professionalism, not punishment i.e. removal of licenses.

> "*I am neutral regarding whether CME CPD should be compulsory because we are adults. The way of supervising postgraduates should be different from that for undergraduates. Being forced to participate may not be acceptable. Instead, incentives may be more appropriate. When you complete a course or seminar, the medical council could award a certificate. If you complete the CME cycle within three years, you can display the certificate in your clinics or print them on your cards. Such incentives are much better than punishments.*" *(A non-specialist in solo private practice)*

A few doctors further mentioned that in order to make CME CPD requirement compulsory, the government would need to provide a suitable (balanced) environment for them to participate i.e. match with "time, location, and target population". In addition, a few participants thought that compulsory CME CPD requirement could be implemented in a phased way starting with medical students since it might be difficult for the older generation to change:

*"Doctors are working for long time and private doctors have to run their businesses. It is important for doctors to have a balance between continuous learning and work. The government should assess how long doctors are working and how many hours can be spared for CME lecture or self-study each week to strike a balance."* (A non-specialist working in the public sector)

## Discussion

The majority of doctors in Hong Kong perceived the need for continuous professional development to update their knowledge due to the advancement in technology and medical knowledge. In general, they agreed that all doctors should be required to participate in CME CPD programmes. CME CPD programmes was the most frequently used method to inform ongoing professional development [18]. In general, doctors in Hong Kong agreed the effectiveness of continuous professional development in updating knowledge and improving practice. The study by UK General Medical Council and Academy of Medical Royal Colleges in 2010 on effectiveness of continuous professional development also showed similar results that the doctors in UK agreed continuous professional development could facilitate changes in treatment and knowledge acquisition [16]. The public in Hong Kong also thought it was important for doctors to participate in CME CPD programmes to keep them updated [10]. However, just 30.7% of non-specialists compared with 65.4% of specialists in the survey agreed it would be desirable to introduce compulsory CME CPD programmes linked to the renewal of licenses. This was probably due to anxiety amongst doctors about such regulatory control, and in particular for non-specialists for whom CME programme is voluntary at this moment. A relatively higher percentage of non-specialists compared with specialists reported encountering barriers to CME CPD participation. The high degree of resistance from non-specialists was probably due to their work nature as community doctors who were mainly self-employed and practice in their own clinics (in "solo practice") (around 70% of all primary care doctors fall into this category), with less flexibility and few incentives to participate in CME CPD programmes. Both in Hong Kong and UK doctors perceived barriers to participating in continuous professional development. There are issues of doctors' long working hours and excess workload in public hospitals in Hong Kong particularly during the influenza season [9]. While UK doctors mentioned cost, work life balance and availability of study leave as the major barriers, Hong Kong doctors further added inconvenience of the schedule and location, and appropriateness of the content of the continuous professional development activities as the major barriers. To address the needs of individual doctors [16], the spheres of practice, personal preferences and learning styles should be considered in deciding the content and processes. Flexibility was frequently cited as an important principle for developing continuous professional development since there was no single or best way of doing it.

Internationally, nearly all jurisdictions studied, except a few states in US and Malaysia (which passed the law in 2012 but not implemented yet), requires all doctors to participate in continuous professional development programmes and links it to the renewal of practicing certificates for licenses e.g. starting from US in 1968, Singapore in 2003, Quebec of Canada in 2007, Australia in 2010. There is substantial international literature providing evidence that participation in continuous professional development is effective in improving professional standards and patients' outcomes [19–23]. Given the international trend in requiring doctors to undertake compulsory continuous professional development to maintain doctors' continuing professional competencies as the basis for patient safety, this could be employed as a driver for encouraging doctors to participate in continuous professional development to acquire

updated knowledge and keep abreast with international trends. Barriers expressed by doctors (in particular those from non-specialists) to participating in continuous professional development also need to be addressed. For example, concerns were made on the format, content and quality of CME CPD programmes which were variable. In order to encourage doctors to take part in CME CPD activities, there should be flexibility in facilitating doctors to participate, such as a convenient location of CME CPD programmes, online CME CPD activities for non-specialists, and most importantly, to integrate CME CPD activities with everyday working practices.

An *"incentives"* type of influence could be considered in encouraging the uptake of continuous professional development [24]. Given the resistance from doctors (in particular for the non-specialists who are mainly primary care doctors in the private sector) to implement compulsory CME CPD programmes linked to the renewal of practicing certificates, a compliance strategy which emphasizes incentives to participate should be considered in the first place rather than a deterrence strategy i.e. discontinuation of licenses. Some other jurisdictions which do not have compulsory continuous professional development requirement have provided incentives to encourage participation. For example, Sweden provides doctors with tax rebates if the doctors join the activities. In Hong Kong, consideration could also be given to require doctors to take part in CME CPD programmes if they wish to enroll in the government funded public private partnership (PPP) programme such as the elderly healthcare voucher scheme. The CME CPD programmes should be relevant to the specific PPP programmes covering aspects of elderly preventive care and chronic disease management in primary care for doctors who wish to enroll in the elderly healthcare voucher scheme.

The incentive compliance strategy and the removal of barriers to participation would facilitate the introduction of compulsory CME CPD programmes for all the doctors in Hong Kong in the medium term as most doctors would have already participated voluntarily. Consequences for non-compliance to compulsory CME CPD programmes can be further examined in Hong Kong to reduce the resistance from doctors. Internationally, sanctions for non-compliance vary throughout the world and range in severity [16]. The most serious one is to link continuous professional development to the renewal of practicing licenses. The medical council/ board has power to remove the license of a doctor if he/ she does not comply with the continuous professional development requirements. Singapore has temporary removal of licenses until the continuous professional development requirements have been fulfilled. In Croatia, doctors may have their licenses to practice revoked till an examination is passed. More often, doctors may be given extra time to fulfill their requirement e.g. up to a year in South Africa. Some jurisdictions e.g. Canada may provide mentors to help doctors fulfill requirements.

Whether a change e.g. compulsory continuous professional development can be implemented depends on various factors such as the historical context, socio-political environment and public interests, and varies from jurisdiction to jurisdiction. Sometimes, government's regulatory objective, incentives for those being regulated and the costs of regulatory failure play a contributory role [25]. There are also political drivers arising from a number of high profile medical incidents arising from fitness to practice which is a catalyst for government's involvement in the reform of medical regulation. Regulatory changes also result from increased patient awareness and expectations about the quality of care leading to public/ patients seeking physician-specific information on the internet, and an increased desire for relevant, accessible and meaningful differentiation of physician competence.

From the findings of the surveys of public and doctors, the levers for change are limited. Therefore, better informed public is needed for an appreciation of the current situation, to advocate changes to the regulatory system which is fair, proportionate, and workable, benefiting both society and doctors.

Based on the educational theories for change, doctors are more motivated to change if individual learning needs, personal motives of professionals, and their personal learning styles are being taking into account [26]. In addition, motivational factors need to be considered. The factors that influenced change processes include curiosity, personal and financial well-being, career planning, wish to improve competence, pressure from patients and colleagues [27]. Doctors need to be provided with clear information of the benefits of change to provide them with the confidence that change is workable and feasible so as to maximize successful implementation. Removal of barriers to participation and incentives to encourage participation is required.

Globally, there is increasing trend to detect and deal with poor performance and to improve quality of care. However, dramatic changes might not be feasible at this moment due to the strong degree of self-regulation in Hong Kong. Any changes will need to be implemented in phase with discussion with the professions and public. Professionalism including continuous professional developing should be encouraged for patient safety.

The postal self-administered questionnaire survey for doctors was conducted among a random sample of all registered doctors in Hong Kong. There might be an over-representation of specialists in our sample. However, the lack of publicly available data on the detailed breakdown of doctors' characteristics did not permit us to conduct weighting on our sample. Instead, we have presented the results by specialists and non-specialists. In addition, doctors who were relatively knowledgeable or had strong views on the topic might be more likely to return the questionnaire.

## Conclusions

With the advances in medical knowledge and the changes in clinical practice, the traditional form of regulation using the *"command and control"* type of influence is complementary by *"incentives-based"* mechanism of influence to drive quality assurance and improvement. The Government could consider using an "incentives-based" mechanism through recognition and incentives to encourage continuous professional development at the beginning, and then moving to a deterrence strategy to link continuous professional development with the renewal of practicing certificates with flexibility for non-specialists. International experience highlights any change in the current monitoring and assessment system should be implemented in phase with more communication, engagement and management of the profession. It should also embrace the principle of professional self-regulation which is accountable to the public. Enabling high levels of professionalism is a key incentive-based strategy which will improve professional standards beyond minimum standards required by registration and licensing. A more active role in influencing, advocacy and support from other stakeholders including public and service providers would also facilitate the efforts to enhance the quality and safety of care. The primary objective is to ensure the doctors are competent and fit to practice to provide high quality care and to support doctors' own professionalism within a quality improvement framework.

## Supporting information

**S1 File.**
(DOCX)

**S2 File.**
(DOCX)

## Acknowledgments

We are most grateful to survey respondents and interviewees for providing us valuable information.

## Author Contributions

**Conceptualization:** Carrie H. K. Yam, Sian M. Griffiths, Eng-Kiong Yeoh.

**Formal analysis:** Carrie H. K. Yam.

**Funding acquisition:** Eng-Kiong Yeoh.

**Methodology:** Carrie H. K. Yam.

**Project administration:** Carrie H. K. Yam.

**Supervision:** Sian M. Griffiths.

**Writing – original draft:** Carrie H. K. Yam.

**Writing – review & editing:** Sian M. Griffiths.

**Writing - Review & Editing:** Eng-Kiong Yeoh.

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
