## [Decision Letter · Decision Letter 0]

1 Oct 2019

PONE-D-19-17445

What helps and hinders doctors in engaging in continuous professional development? An explanatory sequential design

PLOS ONE

Dear Dr Yam,

Thank you for submitting your manuscript to PLOS ONE. After careful consideration, we feel that it has merit but does not fully meet PLOS ONE’s publication criteria as it currently stands. Therefore, we invite you to submit a revised version of the manuscript that addresses the points raised during the review process.

We would appreciate receiving your revised manuscript by Nov 15 2019 11:59PM. To enhance the reproducibility of your results, we recommend that if applicable you deposit your laboratory protocols in protocols.io, where a protocol can be assigned its own identifier (DOI) such that it can be cited independently in the future. For instructions see: http://journals.plos.org/plosone/s/submission-guidelines#loc-laboratory-protocols

We look forward to receiving your revised manuscript.

Kind regards,

Elisa J. F. Houwink, MD, PhD

Academic Editor

PLOS ONE

Journal Requirements:

1. Please clarify in your financial disclosure statement whether any of the funders played any role in the study design, data collection and analysis, decision to publish, or preparation of the manuscript.

3. Please describe and/or include a copy of the interview discussion guide used to interview the participants as a Supporting Information file.

4. Please include a copy of Table (Table Error! Reference source not found.3) which you refer to in your text

5. Please include your tables as part of your main manuscript and remove the individual files. Please note that supplementary tables (should remain/ be uploaded) as separate "supporting information" files

Reviewers' comments:

Reviewer's Responses to Questions

**Comments to the Author**

1. Is the manuscript technically sound, and do the data support the conclusions?

Reviewer #1: No

2. Has the statistical analysis been performed appropriately and rigorously? 

Reviewer #1: No

3. Have the authors made all data underlying the findings in their manuscript fully available?

Reviewer #1: Yes

4. Is the manuscript presented in an intelligible fashion and written in standard English?

Reviewer #1: Yes

5. Review Comments to the Author

Reviewer #1: This paper explores whether doctors in Hong Kong perceive there is a need for continuous professional development and identifies factors that are likely to influence the implementation of compulsory CPD. This is an interesting and practically relevant question. The paper had a large pool of sample and combined quantitative and qualitative methods to answer the research questions. It has clear implications for practice.

The paper would best benefit from the following 2 points:

1. I think there needs to be a stronger motivation for the paper and a stronger grounding in theory.

a. That there is currently no requirement for CPD is not enough as a motivation for why this is needed. I'd suggest giving more background in the HK healthcare system, as many concepts were not clearly explained to an outsider like me (e.g., who are the specialists, what are the pay/salary systems, what about status, what is the cost of medical errors, etc.).

b. There was only a brief discussion of the learning model and the assessment model and it wasn't clear what their roles are in this research. I suggest grounding your paper in one particular theory, writing a more substantive literature review session, and articulating your contributions to theory. As it is, this paper appears very atheoretical.

2. Please clarify your research methods and analyses sections. Right now it is written very generally, thus making it impossible to judge if your methods were rigorous. For example, starting in line 162, what are the questions you asked? How exactly were they developed? What are the response scales? What is an example item? How did you conduct pilot testing? What are the semi-structured interview questions? Please explain all of these details; otherwise, the quality of your research is questionable.

I hope these comments are helpful to the authors in their revision.

6. PLOS authors have the option to publish the peer review history of their article (what does this mean?). If published, this will include your full peer review and any attached files.

Reviewer #1: No

---

## [Author Response · Author response to Decision Letter 0]

15 Nov 2019

Comments from the academic editor

1. Please clarify in your financial disclosure statement whether any of the funders played any role in the study design, data collection and analysis, decision to publish, or preparation of the manuscript.

Responses: The funder had no role in study design, data collection and analysis, decision to publish, or preparation of the manuscript.

Responses: As suggested, we have included a copy of the survey questionnaire as a Supporting Information File.

3. Please describe and/or include a copy of the interview discussion guide used to interview the participants as a Supporting Information file.

Responses: As suggested, we have included a copy of the interview discussion guide as a Supporting Information File.

4. Please include a copy of Table (Table Error! Reference source not found.3) which you refer to in your text.

Responses: We apologize for the error. It refers to Table 3.

5. Please include your tables as part of your main manuscript and remove the individual files. Please note that supplementary tables (should remain/ be uploaded) as separate "supporting information" files

Responses: We have included Tables as part of the manuscript.

 

Comments from the reviewer

1. Is the manuscript technically sound, and do the data support the conclusions?

[The manuscript must describe a technically sound piece of scientific research with data that supports the conclusions. Experiments must have been conducted rigorously, with appropriate controls, replication, and sample sizes. The conclusions must be drawn appropriately based on the data presented.]

Reviewer #1: No

Responses: We have provided more elaboration of the data to support the discussion and conclusion. Methods and analyses part has been revised as suggested. Please see our responses in Q5 item 2 below.

2. Has the statistical analysis been performed appropriately and rigorously? 

Reviewer #1: No

Responses: We have revised the method part. Please see our detailed response in Q5 below.

3. Have the authors made all data underlying the findings in their manuscript fully available?

[The PLOS Data policy requires authors to make all data underlying the findings described in their manuscript fully available without restriction, with rare exception (please refer to the Data Availability Statement in the manuscript PDF file). The data should be provided as part of the manuscript or its supporting information, or deposited to a public repository. For example, in addition to summary statistics, the data points behind means, medians and variance measures should be available. If there are restrictions on publicly sharing data—e.g. participant privacy or use of data from a third party—those must be specified.]

Reviewer #1: Yes

Responses: Thank you.

4. Is the manuscript presented in an intelligible fashion and written in standard English?

[PLOS ONE does not copyedit accepted manuscripts, so the language in submitted articles must be clear, correct, and unambiguous. Any typographical or grammatical errors should be corrected at revision, so please note any specific errors here.]

Reviewer #1: Yes

Responses: Thank you.

5. Review Comments to the Author

[Please use the space provided to explain your answers to the questions above. You may also include additional comments for the author, including concerns about dual publication, research ethics, or publication ethics. (Please upload your review as an attachment if it exceeds 20,000 characters)]

Reviewer #1: This paper explores whether doctors in Hong Kong perceive there is a need for continuous professional development and identifies factors that are likely to influence the implementation of compulsory CPD. This is an interesting and practically relevant question. The paper had a large pool of sample and combined quantitative and qualitative methods to answer the research questions. It has clear implications for practice.

The paper would best benefit from the following 2 points:

1. I think there needs to be a stronger motivation for the paper and a stronger grounding in theory.

a. That there is currently no requirement for CPD is not enough as a motivation for why this is needed. I'd suggest giving more background in the HK healthcare system, as many concepts were not clearly explained to an outsider like me (e.g., who are the specialists, what are the pay/salary systems, what about status, what is the cost of medical errors, etc.).

Responses: We agree that no requirement of CPD is not enough as a motivation for this paper. It is important that there should be continuous learning and regular assessment for doctors to ensure their professional competence. It is also to meet the public expectation for patient safety. However, in Hong Kong policy makers have not been able to mandate participation in CME CPD as a condition for the licence for continuation of practice due to resistance from members of the profession and opposition from members of the regulatory body, the Medical Council of Hong Kong. Similar resistance and opposition from members of the medical profession have also been reported in a number of Asian Jurisdictions when policy makers attempted to mandate CME CPD for the profession. This study seeks to understand the extent and nature of this resistance and identify facilitators which could enable change.

We have provided more information about the medical regulation in the manuscript as suggested.

“Medical doctors in Hong Kong have a high degree of self-regulation. The professional regulatory body, the Medical Council of Hong Kong (MCHK), is dominated by members of the professional with only 8 out of the 32 members being “lay members” who are from outside the profession. To become a medical doctor in Hong Kong, one must enroll in a 6-year accredited medical education programme run by the medical faculties of two universities in Hong Kong. On completion of the medical degree, graduates are given provisional registration by the MCHK to work as interns under supervision in public hospitals under the Hong Kong Hospital Authority for not less than 12 months. After being assessed as having satisfactory performance in the internship training, full registration is given by the MCHK and the doctor may practice medicine independently. Many will however chose to apply for the specialist training with a standard 6-year format for basic and higher training overseen by the Hong Kong Academy of Medicine and its Colleges. However, it is possible to go into independent practice as a non-specialist and not enroll in any further professional training or not pursue continuing medical education and profession development. According to a health manpower survey by Department of Health, 66.7% of doctors work as specialist, with 29.2% in general practice (13). Nearly half (48.9%) of all registered medical practitioners are working in the private sector, while 46.9% work in the public sector. Doctors have relatively a high social status and are paid well in Hong Kong. The salary of a resident in public sector ranges from US$8,700 to US$18,000 per month, which is much higher than the median monthly wage of Hong Kong employees (US$2,200) (14). A resident, after obtaining specialist registration, can be promoted to associate consultant with monthly salary of US$18,200 to US$21,100 after five years of working experience as specialist. Depending on post availability, an associate consultant may be further promoted to a consultant with monthly salary of US$23,500 and above. The salary of doctors in private sector varies; however, in general, it is higher than that of public sector. In some specialties the differential may be several-fold. Though doctors have a relatively higher pay scale in Hong Kong, they are concerned about the long working hours and excess workload in public hospitals particularly during flu season (15).”

b. There was only a brief discussion of the learning model and the assessment model and it wasn't clear what their roles are in this research. I suggest grounding your paper in one particular theory, writing a more substantive literature review session, and articulating your contributions to theory. As it is, this paper appears very atheoretical.

Responses: Thanks. We have strengthened the literature review part. We have linked the learning model and the assessment model to the regulation theory with respect to the regulatory strategies including “command and control” type of influences and the “incentives” type of influences. An “incentives” type of influence could be considered in encouraging the uptake of CME CPD learning given high resistance from doctors. In addition, based on the educational theories for change, doctors are more motivated to change if individual learning needs, personal motives of professionals, and their personal learning styles are being taking into account. With the advances in medical knowledge and the changes in clinical practice, the traditional form of regulation using the “command and control” type of influence is complementary by “incentives-based” mechanism of influence to drive quality assurance and improvement.

2. Please clarify your research methods and analyses sections. Right now it is written very generally, thus making it impossible to judge if your methods were rigorous. For example, starting in line 162, what are the questions you asked? How exactly were they developed? What are the response scales? What is an example item? How did you conduct pilot testing? What are the semi-structured interview questions? Please explain all of these details; otherwise, the quality of your research is questionable.

I hope these comments are helpful to the authors in their revision.

Responses: Thank you for your comments. We have included details in the methods and analyses part as suggested.

“The questions were drafted taking reference to literature about doctors’ views towards medical regulation and continuous professional development (15-17). The questionnaire had been commented on and revised by experts, and pilot-tested before implementation in the main study. The questionnaire focused on the perceived needs for, and barriers to, continuous professional development (Supporting Document 1). There were four questions relating to life-long learning needs e.g. whether they recognized the need to regularly update medical knowledge. After understanding their needs, we asked whether the respondents actually participated in any self-learning and CME CPD activities in the past one year, and if there were any barriers encountered to CME CPD learning. We further explored their perception on the usefulness of continuous professional development e.g. whether taking part in CME CPD can improve their practical skills or patient outcome; and also whether they agree that this should be a compulsory requirement for all doctors to participate in CME CPD. The questions were presented in a 4-point Likert scale to show the extent of agreement of the participants towards the questions. Pilot testing was conducted among 10 medical doctors in face-to-face interviews and amendments to the wordings were made to the questionnaire based on the feedback received.”

Semi-structured interviews questions were provided as supporting document 2, and elaborated as follows:

“For example, for those who did not agree that all doctors should participate in CME CPD, we explored their reasons why they thought not every doctor should engage in CME CPD, and to suggest whether there were any other ways they could keep medical knowledge up to date. We also asked participants for their input for mechanisms to reduce the barriers to CME CPD learning, and for ways to encourage doctors to participation in CME CPD. The preferred type of activity and format of CME CPD were also explored.”

---

## [Editor Report · Decision Letter 1]

12 Jun 2020

PONE-D-19-17445R1

What helps and hinders doctors in engaging in continuous professional development? An explanatory sequential design

PLOS ONE

Dear Dr Yam,

Thank you for submitting your manuscript to PLOS ONE. After careful consideration, we feel that it has merit but does not fully meet PLOS ONE’s publication criteria as it currently stands. Therefore, we invite you to submit a revised version of the manuscript that addresses the points raised during the review process.

Comments for your attention are provided at the end of the email.

We would appreciate receiving your revised manuscript by Jul 27 2020 11:59PM. To enhance the reproducibility of your results, we recommend that if applicable you deposit your laboratory protocols in protocols.io, where a protocol can be assigned its own identifier (DOI) such that it can be cited independently in the future. For instructions see: http://journals.plos.org/plosone/s/submission-guidelines#loc-laboratory-protocols

We look forward to receiving your revised manuscript.

Kind regards,

Jenny Wilkinson, PhD

Academic Editor

PLOS ONE

Additional Editor Comments (if provided):

Thank you for your responses and manuscript revisions. These have addressed most of the comments with some additional comments arising from the revised text (below). In addition while overall the language is acceptable however it would benefit from careful proofing to address grammatical and language issues.

1. The Abstract should be shortened to a maximum of 300 words

2. Line 27 Introduces public expectations as a driver of the work however it is not clear what this expectation is and how it fits with CME and CPD; public expectation also do not seem to be a focus on the Introduction to the work

3. The Introduction refers to both CME and CPD but does not differentiate between the two or the rationale for use of the terms singularly or in combination in this work. In many jurisdictions CPD is used as the umbrella term encompassing all post licensure learning needed to maintain clinical skills and knowledge. As the term used in MCHK is CME it is also unclear what respondents understood CME CPD to be as distinct from CME or CPD, or whether they understood the three terms to be synonymous with each other.

4. A new section of text has been added between lines 208-218 discussing salaries however it is unclear how this relates to CME/CPD.

---

## [Author Response · Author response to Decision Letter 1]

24 Jul 2020

Comments 

Thank you for your responses and manuscript revisions. These have addressed most of the comments with some additional comments arising from the revised text (below). In addition while overall the language is acceptable however it would benefit from careful proofing to address grammatical and language issues.

Responses: 

Thank you for your comments. We have proof-read the manuscript to correct the grammatical and language issues.

1. The Abstract should be shortened to a maximum of 300 words

Responses: 

We have shortened the abstract to 300 words.

2. Line 27 Introduces public expectations as a driver of the work however it is not clear what this expectation is and how it fits with CME and CPD; public expectation also do not seem to be a focus on the Introduction to the work.

Responses: 

Thank you for your comments. In Hong Kong, there is no mandated, structured and ongoing mechanism to assess all doctors’ performance and ensure their competence in Hong Kong. However, the public expect doctors to stay up-to-date and improve the safety and quality of healthcare services. Taking part in continuous professional development is one of the ways to maintain and improve practice of doctors. We agree public expectation is not the main focus for this work. Engaging doctors in continuous professional development is important for their professional development as well as patient safety due to advances in the medical technology and changes in clinical practice. We have revised this paragraph to make it clearer.

3. The Introduction refers to both CME and CPD but does not differentiate between the two or the rationale for use of the terms singularly or in combination in this work. In many jurisdictions CPD is used as the umbrella term encompassing all post licensure learning needed to maintain clinical skills and knowledge. As the term used in MCHK is CME it is also unclear what respondents understood CME CPD to be as distinct from CME or CPD, or whether they understood the three terms to be synonymous with each other.

Responses: 

Thank you for your comments and apologize for not changing the terms used in the Hong Kong context. Many jurisdictions e.g. US and UK usually use continuous professional development (CPD) as the collective term to describe all post-licensure learning to maintain professional competence. However, in our local context, the Hong Kong Academy of Medicine has statutory responsibility for accreditation and training of all specialists. Continuous professional development for specialists is differentiated into components of Continuing Medical Education (CME) and Continuous Professional Development (CPD) with guidelines on the types and hours of programmes required for each of these 2 components required for renewal of specialist status. “CME CPD programmes” is the name which is used by the Hong Kong Academy of Medicine. The Hong Kong Medical Council is the statutory authority that licenses all doctors and registers specialists. Neither CME nor CPD is mandated for renewal of licenses. However, to encourage continuous medical education for non-specialists, the Medical Council of Hong Kong has launched a voluntary CME programme, and issues a “CME Certificate” to those doctors who fulfill the requirements. We have revised the text in using “CME CPD programmes” specifically for Hong Kong’s context, while we use the collective term “continuous professional development” for general purpose.

4. A new section of text has been added between lines 208-218 discussing salaries however it is unclear how this relates to CME/CPD.

Responses: 

Thank you for your comments. This section is added in response to the reviewer’s comments in our first round of revision. The reviewer suggested to provide more background information of Hong Kong’s healthcare system (e.g. who are the specialists and their status and pay/ salary systems). This background information could facilitate the reader understand Hong Kong’s system, and appreciate the relatively high social status of doctors in Hong Kong. We have kept this information but have abbreviated the description.

---

## [Editor Report · Decision Letter 2]

31 Jul 2020

What helps and hinders doctors in engaging in continuous professional development? An explanatory sequential design

PONE-D-19-17445R2

Dear Dr. Yam,

We’re pleased to inform you that your manuscript has been judged scientifically suitable for publication and will be formally accepted for publication once it meets all outstanding technical requirements.

Kind regards,

Jenny Wilkinson, PhD

Academic Editor

PLOS ONE

Additional Editor Comments (optional):

Thank you for your revisions to address review comments; these have been satisfactorily addressed.
---

## [Editor Report · Acceptance letter]

4 Aug 2020

PONE-D-19-17445R2 

What helps and hinders doctors in engaging in continuous professional development? An explanatory sequential design 

Dear Dr. Yam:

I'm pleased to inform you that your manuscript has been deemed suitable for publication in PLOS ONE. Congratulations! Your manuscript is now with our production department. 

Kind regards, 

on behalf of

Prof. Jenny Wilkinson 

Academic Editor

PLOS ONE